



Formation mechanisms of atmospheric nitrate and sulfate during the
winter haze pollution periods in Beijing: gas-phase, heterogeneous
and aqueous-phase chemistry
**Pengfei Liu**[1, 2, 3, 5]**, Can Ye**[1, 3]**, Chaoyang Xue**[1, 3]**, Chenglong Zhang**[1, 2, 3]**, Yujing Mu**[1, 2, 3, 4]**, Xu**
**Sun**[1, 6]
[1] Research Center for Eco-Environmental Sciences, Chinese Academy of Sciences, Beijing, 100085, China.
[2] Center for Excellence in Urban Atmospheric Environment, Institute of Urban Environment, Chinese Academy of
Sciences, Xiamen, 361021, China.
[3] University of Chinese Academy of Sciences, Beijing, 100049, China.
[4] National Engineering Laboratory for VOCs Pollution Control Material & Technology, University of Chinese
Academy of Sciences, Beijing, 100049, China.
[5] Key Laboratory of Atmospheric Chemistry, China Meteorological Administration, Beijing, 100081, China.
[6] Beijing Urban Ecosystem Research Station, Beijing, 100085, China.
**Correspondence: Yujing Mu** (yjmu@rcees.ac.cn)
**Abstract**
A vast area in China is currently going through severe haze episodes with drastically elevated
concentrations of $PM_{2.5}$ in winter. Nitrate and sulfate are main constituents of $PM_{2.5}$ but their
formations via $NO_2$ and $SO_2$ oxidation are still not comprehensively understood, especially under
different pollution or atmospheric relative humidity (RH) conditions. To elucidate formation
pathways of nitrate and sulfate in different polluted cases, hourly samples of $PM_{2.5}$ were collected
continuously in Beijing during the wintertime of 2016. Three serious pollution cases were
identified reasonably during the sampling period and the secondary formations of nitrate and
sulfate were found to make a dominant contribution to atmospheric $PM_{2.5}$ under the relatively high
RH condition. The significant correlation between NOR and $NO_2 \times O_3$ during the nighttime under
the RH≥60% condition indicated that the heterogeneous hydrolysis of $N_2O_5$ involving aerosol
liquid water was responsible for the nocturnal formation of nitrate at the extremely high RH levels.
The more coincident trend of NOR and HONO × DR (direct radiation) × $NO_2$ than Dust × $NO_2$





during the daytime under the 30%<RH<60% condition provided convincing evidence that the
gas-phase reaction of $NO_2$ with OH played a pivotal role in the diurnal formation of nitrate at
moderate RH levels. The extremely high mean values of SOR during the whole day under the
RH≥60% condition could be ascribed to the evident contribution of $SO_2$ aqueous-phase oxidation
to the formation of sulfate during the severe pollution episodes. Based on the parameters measured
in this study and the known sulfate production rate calculation method, the oxidation pathway of
$H_2O_2$ rather than $NO_2$ was found to contribute greatly to the aqueous-phase formation of sulfate.
**1. Introduction**

In recent years, severe haze has occurred frequently in Beijing as well as the North China

Plain (NCP) during the wintertime, which has aroused great attention from the public due to its
adverse impact on atmospheric visibility, air quality and human health (Chan and Yao,
2008;Zhang et al., 2012;Zhang et al., 2015).

To mitigate the severe haze pollution situations, a series of regulatory measures for primary

pollution sources have been implemented by the Chinese government. For example, coal
combustion for heating in winter has gradually been replaced with electricity and natural gas in
the NCP, coal-fired power plants have been strictly required to install flue-gas denitration and
desulfurization systems (Chen et al., 2014), the stricter control measures such as terminating
production in industries and construction as well as the odd and even number rule for vehicles
have been performed in megacities during the period of the red alert for haze and so on. These
actions have made tremendous effects to decline pollution levels of primary pollutants including
$PM_{2.5}$ (fine particulate matter with an aerodynamic diameter less than 2.5 μm) in recent years (Li
et al., 2019). However, the serious pollution events still occurred in many areas of




Beijing-Tianjin-Hebei (BTH) region in December 2016 and January 2017 (Li et al., 2019). It has
been acknowledged that the severe haze pollution is mainly ascribed to stagnant meteorological
conditions with high atmospheric relative humidity (RH) and low mixed boundary layer height,
strong emissions of primary gaseous pollutants and rapid formation of secondary inorganic
aerosols (SIAs, the sum of sulfate, nitrate and ammonium), especially sulfate and nitrate (Cheng et
al., 2016;Guo et al., 2014;Huang et al., 2014). Some studies suggested that the contribution of
SIAs to $PM_{2.5}$ was higher than 50 % during the most serious haze days (Quan et al., 2014;Xu et al.,
2017;Zheng et al., 2015a).

Generally, atmospheric sulfate and nitrate are formed through the oxidations of the precursor

gases ($SO_2$ and $NO_2$) by oxidants (e.g. OH radical, $O_3$) via gas-phase, heterogeneous and
aqueous-phase reactions (Ravishankara, 1997;Wang et al., 2013;Yang et al., 2015). It should be
noted that the recent study proposed the remarkable emissions of primary sulfate from residential
coal combustion with the sulfur contents of coal in range of 0.81-1.88 % in Xi'an (Dai et al.,
2019), but the primary emissions of sulfate could be neglected due to the extremely low sulfur
content of coal (0.26-0.34 %) used prevailingly in the NCP (Du et al., 2016;Li et al., 2016).
Atmospheric RH is a key factor that facilitates the SIAs formation and aggravates the haze
pollution (Wu et al., 2019), and hence the secondary formations of sulfate and nitrate are simply
considered to be mainly via gas-phase reaction at relatively low atmospheric RH levels (RH<30 %)
and heterogeneous reactions and aqueous-phase reactions at relatively high atmospheric RH levels
(RH>60 %) (Li et al., 2017). However, their formation mechanisms at different atmospheric RH
levels still remain controversial and unclear (Cheng et al., 2016;Ge et al., 2017;Guo et al., 2017;Li
et al., 2018;Liu et al., 2017a;Wang et al., 2016;Yang et al., 2017). For example, the recent studies



proposed that atmospheric $SO_2$ oxidation by $NO_2$ dissolved in aqueous aerosol phases under the
extremely high atmospheric RH conditions played a dominant role in sulfate formation under
almost neutral aerosol solutions (a pH range of 5.4-7.0) during the serious pollution periods
(Cheng et al., 2016;Wang et al., 2018a;Wang et al., 2016). However, Liu et al. (2017a) and Guo et
al. (2017) found that the aerosol pH estimated by ISORROPIA-II model was moderately acidic (a
pH range of 3.0-4.9) and thus the pathway of $SO_2$ aqueous-phase oxidation by dissolved $NO_2$ was
unimportant during severe haze events in China. Additionally, although the pathway of $N_2O_5$
heterogeneous hydrolysis has been recognized as being responsible for the nocturnal formation of
$NO_3^-$ under relatively high atmospheric RH conditions (Tham et al., 2018;Wang et al.,
2018b;Wang et al., 2018c), the effects of $NO_2$ gas-phase chemistry and $NO_2$ heterogeneous
chemistry on the diurnal formation of $NO_3^-$ under moderate atmospheric RH conditions
(30 %<RH<60 %) have not yet been understood. Therefore, measurements of the species in $PM_{2.5}$
in different polluted cases during the wintertime are urgently needed to elucidate formation
pathways of sulfate and nitrate.

In this study, hourly filter samples of $PM_{2.5}$ were collected continuously in Beijing during the

wintertime of 2016, and the pollution characteristics and formation mechanisms of sulfate and
nitrate in the $PM_{2.5}$ samples were investigated comprehensively under different atmospheric RH
conditions in relation to gas-phase, heterogeneous and aqueous-phase chemistry.
**2. Materials and Methods**
**2.1. Sampling and analysis**

The sampling site was chosen on the rooftop (around 25 m above the ground) of a six-story

building in Research Center for Eco-Environmental Sciences, Chinese Academy of Sciences


(RCEES, CAS), which was located in the northwest of Beijing and had been described in detail by
our previous studies (Liu et al., 2016a;Liu et al., 2017b). The location of the sampling site
(40°00′29.85″ N, 116°20′29.71″ E) is presented in Figure S1. Hourly $PM_{2.5}$ samples were collected
on prebaked quartz fiber filters (90mm, Munktell) from January 7th to 23th of 2016 by
median-volume samplers (Laoying-2030) with a flow rate of 100 L min$^{-1}$. Water-soluble ions
(WSI), including $Na^+$, $NH_4^+$, $Mg^{2+}$, $Ca^{2+}$, $K^+$, $Cl^-$, $NO_2^-$, $NO_3^-$ and $SO_4^{2-}$, as well as carbon
components including organic carbon (OC) and element carbon (EC) in the filter samples were
analyzed by ion chromatography (Wayeal IC6200) and thermal optical carbon analyzer
(DRI-2001A), respectively (Liu et al., 2017b). Analysis relevant for quality assurance & quality
control (QA/QC) was presented in detail in section M1 of the Supplementary Information (SIs).
Atmospheric $H_2O_2$ and HONO were monitored by AL2021-$H_2O_2$ monitor (AERO laser, Germany)
and a set of double-wall glass stripping coil sampler coupled with ion chromatography (SC-IC),
respectively (Ye et al., 2018;Xue et al., 2019a;Xue et al., 2019b). More details about the
measurements of $H_2O_2$ and HONO were ascribed in section M2 of the SIs. Meteorological data,
including wind speed, wind direction, ambient temperature and RH, as well as air quality index
(AQI) derived by $PM_{2.5}$, $SO_2$, $NO_x$, CO and $O_3$ were obtained from Beijing urban ecosystem
research station in RCEES, CAS (http://www.bjurban.rcees.cas.cn/).
**2.2. Aerosol liquid water contents and pH prediction by ISORROPIA-II model**
The ISORROPIA-II model was employed to calculate the equilibrium composition for
$Na^+$-$K^+$-$Ca^{2+}$-$Mg^{2+}$-$NH_4^+$-$Cl^-$-$NO_3^-$-$SO_4^{2-}$-$H_2O$ aerosol system, which is widely used in regional
and global atmospheric models and has been successfully applied in numerous studies for
predicting the physical state and composition of atmospheric inorganic aerosols (Fountoukis and



Nenes, 2007;Guo et al., 2015;Shi et al., 2017). It can be used in two modes: forward mode and
reverse mode. Forward mode calculates the equilibrium partitioning given the total concentrations
of gas and aerosol species, whereas reverse mode involves predicting the thermodynamic
compositions based only on the concentrations of aerosol components. Forward mode was
adopted in this study because reverse mode calculations have been verified to be not suitable to
characterize aerosol acidity (Guo et al., 2015;Hennigan et al., 2015;Murphy et al., 2017;Pathak et
al., 2004;Weber et al., 2016). The ISORROPIA-II model is available in "metastable" or "solid +
liquid" state solutions. Considering the relatively high RH during the sampling period, the
metastable state solution was selected in this study due to its better performance than the latter
(Bougiatioti et al., 2016;Guo et al., 2015;Liu et al., 2017a;Weber et al., 2016). Additionally,
although the gaseous $HNO_3$, $H_2SO_4$, HCl and $NH_3$ were not measured in this study, gas-phase
input with the exception of $NH_3$ has an insignificant impact on the aerosol liquid water contents
(ALWC) and pH calculation due to the lower concentrations of $HNO_3$, $H_2SO_4$ and HCl than $NH_3$
in the atmosphere (Ding et al., 2019;Guo et al., 2017). Based on the long-term measurement in the
winter of Beijing, an empirical equation between $NO_x$ and $NH_3$ concentrations was derived from
the previous study (Meng et al., 2011), that is, $NH_3$ (ppb) = 0.34 × $NO_x$ (ppb) + 0.63, which was
employed for estimating the $NH_3$ concentration in this study. The predicted daily average
concentrations of $NH_3$ varied from 3.3 μg m$^{-3}$ to 36.9 μg m$^{-3}$, with a mean value of 16.6 μg m$^{-3}$
and a median value of 14.6 μg m$^{-3}$, which were in line with those (7.6-38.1 μg m$^{-3}$, 18.2 μg m$^{-3}$
and 16.2 μg m$^{-3}$ for the daily average concentrations, the mean value and the median value of $NH_3$,
respectively) during the winter of 2013 in Beijing in the previous study (Zhao et al., 2016).

Then, the aerosol pH could be calculated by the following equation:



$$pH = -log_{10}\frac{1000 \times H^+}{W}$$
where $H^+$ (µg m⁻³) and $W$ (µg m⁻³) are the equilibrium particle hydrogen ion concentration
and aerosol water contents, respectively, both of which could be output from ISORROPIA-II.
**2.3. Production of sulfate in aqueous-phase reactions**

The previous studies showed that there were six pathways of the aqueous-phase oxidation of

$SO_2$ to the production of sulfate, i.e. $H_2O_2$ oxidation, $O_3$ oxidation, $NO_2$ oxidation, transition metal
ions (TMI) + $O_2$ oxidation, methyl hydrogen peroxide (MHP) oxidation and peroxyacetic acid
(PAA) oxidation (Cheng et al., 2016;Zheng et al., 2015a). Because some TMIs, such as Ti(III),
V(III), Cr(III), Co(II), Ni(II), Cu(II) and Zn(II), displayed much less catalytic activities (Cheng et
al., 2016), only Fe(III) and Mn(II) were considered in this study. In addition, due to the extremely
low concentrations of MHP and PAA in the atmosphere, their contributions to the production of
sulfate could be ignored (Zheng et al., 2015a). To investigate the formation mechanism of sulfate
during the serious pollution episodes, the contributions of $O_3$, $H_2O_2$, $NO_2$ and Fe(III) + Mn(II) to
the production of sulfate in aqueous-phase reactions were calculated by the formulas as follows
(Cheng et al., 2016;Ibusuki and Takeuchi, 1987;Seinfeld and Pandis, 2006):
$$-(\frac{d[S(IV)]}{dt})_{O_3} = (k_0[SO_2 \cdot H_2O] + k_1[HSO_3^-] + k_2[SO_3^{2-}])[O_{3(aq)}] \qquad \text{(R1)}$$
$$-(\frac{d[S(IV)]}{dt})_{H_2O_2} = \frac{k_3[H^+][HSO_3^-][H_2O_{2(aq)}]}{1+K[H^+]} \qquad \text{(R2)}$$
$$-(\frac{d[S(IV)]}{dt})_{Fe(III)+Mn(II)} = k_4[H^+]^a[Mn(II)][Fe(III)][S(IV)] \qquad \text{(R3)}$$
$$-(\frac{d[S(IV)]}{dt})_{NO_2} = k_5[NO_{2(aq)}][S(IV)] \qquad \text{(R4)}$$
where $k_0 = 2.4 \times 10^4$ M⁻¹ s⁻¹, $k_1 = 3.7 \times 10^5$ M⁻¹ s⁻¹, $k_2 = 1.5 \times 10^9$ M⁻¹ s⁻¹, $k_3 = 7.45 \times 10^7$ M⁻¹ s⁻¹, $K =$
13 M⁻¹, $k_4 = 3.72 \times 10^7$ M⁻¹ s⁻¹, $a = -0.74$ (pH≤4.2) or $k_4 = 2.51 \times 10^{13}$ M⁻¹ s⁻¹, $a = 0.67$ (pH>4.2), and
$k_5 = (1.24-1.67) \times 10^7$ M⁻¹ s⁻¹ (5.3≤pH≤8.7, the linear interpolated values were used for pH between





5.3 and 8.7) at 298 K (Clifton et al., 1988); $[O_{3(aq)}]$, $[H_2O_{2(aq)}]$ and $[NO_{2(aq)}]$ could be calculated by
the Henry's constants which are $1.1 \times 10^{-2}$ M atm$^{-1}$, $1.0 \times 10^5$ M atm$^{-1}$ and $1.0 \times 10^{-2}$ M atm$^{-1}$ at 298 K
for $O_3$, $H_2O_2$ and $NO_2$ respectively (Seinfeld and Pandis, 2006). As for [Fe(III)] and [Mn(II)], their
concentrations entirely depended on the values of pH due to the precipitation equilibriums of
$Fe(OH)_3$ and $Mn(OH)_2$ (Graedel and Weschler, 1981). Considering the aqueous-phase ionization
equilibrium of $SO_2$, the Henry's constants of $HSO_3^-$, $SO_3^{2-}$ and S(IV) could be expressed by the
equations as follows (Seinfeld and Pandis, 2006):
$$H^*_{HSO_3^-} = H_{SO_2} \frac{K_{S1}}{[H^+]} \tag{R5}$$

$$H^*_{SO_3^{2-}} = H_{SO_2} \frac{K_{S1} K_{S2}}{[H^+]^2} \tag{R6}$$

$$H^*_{S(IV)} = H_{SO_2} \left(1 + \frac{K_{S1}}{[H^+]} + \frac{K_{S1} K_{S2}}{[H^+]^2}\right) \tag{R7}$$

where $H_{SO2} = 1.23$ M atm$^{-1}$, $K_{S1} = 1.3 \times 10^{-2}$ M and $K_{S2} = 6.6 \times 10^{-8}$ M at 298 K. In addition, all
of rate constants (k), Henry's constants (H) and ionization constants (K) are evidently influenced
on the ambient temperature and are calibrated by the formulas as follows (Seinfeld and Pandis,

2006):

$$k(T) = k(T_0) e^{\left[-\frac{E}{R}\left(\frac{1}{T} - \frac{1}{T_0}\right)\right]} \tag{R8}$$

$$H(T) = H(T_0) e^{\left[-\frac{\Delta H}{R}\left(\frac{1}{T} - \frac{1}{T_0}\right)\right]} \tag{R9}$$

$$K(T) = K(T_0) e^{\left[-\frac{E}{R}\left(\frac{1}{T} - \frac{1}{T_0}\right)\right]} \tag{R10}$$

where T is the ambient temperature, $T_0$=298 K, both E/R and ΔH/R varied in the different
equations and their values could be found in Cheng et al., (2016).
Furthermore, mass transport was also considered for multiphase reactions in different
medium and across the interface in section M3 of the SIs.
**3. Results and Discussion**





**3.1. Variation characteristics of the species in PM$_{2.5}$ and typical gaseous pollutants**
The concentrations of the species in PM$_{2.5}$ and typical gaseous pollutants including NO$_x$, SO$_2$,
O$_3$, HONO and H$_2$O$_2$ as well as atmospheric RH are shown in Figure 1. The meteorological
parameters (wind speed, wind direction, ambient temperature and direct radiation (DR)) as well as
the concentrations of PM$_{2.5}$ are displayed in Figure S2. During the sampling period, the
concentrations of the species in PM$_{2.5}$ and typical gaseous pollutants varied similarly on a
timescale of hours with a distinct periodic cycle of 3-4 days, suggesting that meteorological
conditions played a vital role in accumulation and dispersion of atmospheric pollutants (Xu et al.,
2011;Zheng et al., 2015b). For example, the relatively high levels of PM$_{2.5}$ (>100 μg m$^{-3}$) usually
occurred under the relatively stable meteorological conditions with the low south wind speed (<2
m s$^{-1}$) and the high RH (>60 %) which favored the accumulation of pollutants. Besides
meteorological conditions, the extremely high concentrations of the species in PM$_{2.5}$ might be
mainly ascribed to strong emissions of primary pollutants and rapid formation of secondary
aerosols during the wintertime in Beijing.
The average concentrations of the species in PM$_{2.5}$ and typical gaseous pollutants during
clean or slightly polluted (C&SP) episodes (PM$_{2.5}$<75 μg m$^{-3}$), during polluted or heavy polluted
(P&HP) episodes (PM$_{2.5}$≥75 μg m$^{-3}$) and during the whole sampling period are illustrated in Table
1. It is evident that the average concentrations of NO$_3^-$, SO$_4^{2-}$, NH$_4^+$, OC and EC during P&HP
episodes were about a factor of 5.0, 4.1, 6.1, 3.6 and 3.2 greater than those during C&SP episodes,
respectively, indicating that the formations of SIAs were more efficient compared to other species
in PM$_{2.5}$ during the serious pollution episodes. Given that the average concentrations of gaseous
precursors (NO$_2$ and SO$_2$) during P&HP episodes were approximately a factor of 2.0-2.2 greater
than those during C&SP episodes, the obviously higher elevation of $NO_3^-$ and $SO_4^{2-}$ implied that
the oxidations of $NO_2$ and $SO_2$ by the major atmospheric oxidizing agents (OH radicals, $O_3$ and
$H_2O_2$ etc.) might be greatly accelerated due to the relatively high concentrations of oxidants and
atmospheric RH during the serious pollution episodes (Figure 1). The average concentration of
$H_2O_2$ was found to be a factor of 1.7 greater during P&HP episodes than during C&SP episodes,
indicating that atmospheric $H_2O_2$ might contribute to the formation of SIAs especially sulfate
during the serious pollution episodes with high atmospheric RH, which will be discussed in Sect.
3.3.2. However, the obvious decrease in $O_3$ average concentration was observed during P&HP
episodes compared to C&SP episodes, which was mainly attributed to the relatively weak solar
radiation and the titration of NO during the serious pollution episodes (Ye et al., 2018). In addition,
the evidently higher average concentration of HONO during P&HP episodes than during C&SP
episodes might be also due to the relatively weak solar radiation as well as the heterogeneous
reaction of $NO_2$ on particle surfaces during the serious pollution episodes (Tong et al., 2016;Wang
et al., 2017).
**3.2. Three serious pollution cases during the sampling period**

Based on the transition from the clean to polluted periods, three haze cases were identified

during the sampling period (Figure 1 and Figure S2): from 13:00 on January 8th to 1:00 on January
11th (Case 1), from 14:00 on January 14th to 7:00 on January 17th (Case 2), and from 8:00 on
January 19th to 2:00 on January 22nd (Case 3). The serious pollution duration in the three cases
could last 1-3 days due to the differences of their formation mechanisms.

In Case 1, the variation trends of the concentrations of the species in $PM_{2.5}$, $NO_x$, $SO_2$,

HONO and $H_2O_2$ were almost identical and exhibited three pollution peaks at night (Figure 1),



which might be ascribed to the possibility that the decrease of nocturnal mixed boundary layer
accelerated the pollutant accumulation (Bei et al., 2017;Zhong et al., 2019). Considering the
relatively low RH (15-40 %) and wind speeds ($<2$ m s$^{-1}$) in Case 1 (Figure S2), primary emissions
around the sampling site were suspected to be a dominant source for the increase in the $PM_{2.5}$
concentrations. Further evidence is that the correlation between the concentrations of $PM_{2.5}$ and
CO is better in Case 1 ($R^2$=0.55) than in Case 2 and Case 3 ($R^2$=0.20~0.52) (Figure S3). Identical
to Case 1, three obvious pollution peaks were also observed in Case 2 (Figure 1). The variation
trends of the concentrations of the species in $PM_{2.5}$ and typical gaseous pollutants at the first peak
in Case 2 were found to be similar with those in Case 1, which were mainly attributed to their
similar formation mechanism. However, the evident decreases in $NO_x$ and $SO_2$ were observed
when the concentrations of the species in $PM_{2.5}$ were increasing and the atmospheric oxidation
pollutant (e.g. $H_2O_2$) concentration peaks were prior to others at the last two peaks in Case 2,
suggesting that secondary formation from gaseous precursors might be dominant for $PM_{2.5}$
pollution. The relatively high RH (50-80 %) and the low south wind speeds ($<2$ m s$^{-1}$) in Case 2
(Figure S2) provided further evidence for the above speculation. In contrast to Case 1 and Case 2,
the relatively high south wind speeds ($>3$ m s$^{-1}$) (Figure S2) with the concentrations of the species
in $PM_{2.5}$ and typical gaseous pollutants increasing slowly (Figure 1) at the beginning of Case 3
indicated that regional transportation might be responsible for the atmospheric species.
Subsequently, the concentrations of the species in $PM_{2.5}$ remained relatively high when the
atmospheric RH lasted more than 60 %, implying that secondary formation from gaseous
precursors dominated $PM_{2.5}$ pollution during the late period of Case 3.
The average mass proportions of the species in $PM_{2.5}$ in the three cases are illustrated in


Figure S4, the proportions of the primary species such as EC (10-13 %), Cl⁻ (6-7 %) and Na⁺ (4 %)
in the three cases were almost identical, indicating that primary particle emissions were relatively
stable during the sampling period. However, the proportions of SIA in Case 2 (42 %) and Case 3
(38 %) were conspicuously greater than that in Case 1 (28 %), further confirming that secondary
formation of inorganic ions (e.g. nitrate, sulfate) made a significant contribution to atmospheric
$PM_{2.5}$ in Case 2 and Case 3.
**3.3. Formation mechanism of nitrate and sulfate during serious pollution episodes**
As for nitrate and sulfate in the three cases, the highest mass proportion (18 %) of nitrate was
observed in Case 2, whereas the highest mass proportion (15 %) of sulfate was found in Case 3
(Figure S4). Although the concentrations of $SO_2$ were about a factor of 5 lower than the
concentrations of $NO_2$ in both Case 2 and Case 3 (Figure 1), the extremely high proportion of
sulfate in Case 3 might be ascribed to the long-lasting plateau of RH (Figure S2) because the
aqueous-phase reaction could accelerate the conversion of $SO_2$ to $SO_4^{2-}$. To further investigate the
pollution characteristics of nitrate and sulfate during the serious pollution episodes, the relations
between NOR (NOR = $NO_3^-$ / ($NO_3^-$+$NO_x$)) as well as SOR (SOR = $SO_4^{2-}$ / ($SO_4^{2-}$+$SO_2$)) and RH
are shown in Figure 2. There were obvious differences in the variations of NOR and SOR under
different atmospheric RH conditions. The variation trends of NOR and SOR almost stayed the
same when atmospheric RH was below 30 %, and then simultaneously increased with atmospheric
RH in the range of 30-60 %. The enhanced gas-phase reaction and the heterogeneous reaction
involving aerosol liquid water might make a remarkable contribution to the elevation of NOR and
SOR, respectively, which were further discussed in the following section. Subsequently, the
variation trend of NOR slowly decreased whereas the variation trend of SOR significantly





increased when atmospheric RH was above 60 %. The reduction of NOR might be due to the
deliquescence of nitrate at atmospheric RH around 60 % (Kuang et al., 2016;Liu et al., 2016b;Xue
et al., 2014), while the elevation of SOR revealed the dominant contribution of the aqueous-phase
reaction to the formation of sulfate.
**3.3.1. Formation mechanism of nitrate**
Atmospheric nitrate is considered to be mainly from $NO_2$ oxidation by OH radical in the gas
phase, heterogeneous uptake of $NO_2$ on the surface of particles and heterogeneous hydrolysis of
$N_2O_5$ on wet aerosols or chloride-containing aerosols (He et al., 2014;He et al., 2018;Nie et al.,
2014;Ravishankara, 1997;Wang et al., 2018b). Since atmospheric $N_2O_5$ is usually produced by the
reaction of $NO_3$ radical with $NO_2$ as well as both $NO_3$ radical and $N_2O_5$ are easily photolytic
during the daytime, the heterogeneous hydrolysis of $N_2O_5$ is a nighttime pathway for the
formation of atmospheric nitrate (He et al., 2018;Wang et al., 2018b). As shown in Figure 3a, the
mean values of NOR during the nighttime remarkably elevated with atmospheric RH increasing,
the disproportionation of $NO_2$ and the heterogeneous hydrolysis of $N_2O_5$ involving aerosol liquid
water were suspected to dominate the nocturnal formation of nitrate under high RH conditions
during the sampling period (Ma et al., 2017;Wang et al., 2018b;Li et al., 2018). However, the
productions of HONO and nitrate should be equal through the disproportionation of $NO_2$ (Ma et
al., 2017), which could not explain the wide gaps between the average concentrations of HONO
(about 6.5 μg m$^{-3}$) and nitrate (about 20.1 μg m$^{-3}$) observed at the nighttime under high RH
conditions during the sampling period. Thus, the disproportionation of $NO_2$ made insignificant
contribution to the nocturnal formation of nitrate under high RH conditions. Considering that the
formation of atmospheric $NO_3$ radical is mainly via the oxidation of $NO_2$ by $O_3$, the heterogeneous



hydrolysis of $N_2O_5$ occurs only at high $O_3$ and $NO_2$ levels during the nighttime (He et al.,
2018;Wang et al., 2018b). Therefore, the correlation between $NO_2 \times O_3$ and NOR can represent
roughly the contribution of the heterogeneous hydrolysis of $N_2O_5$ to atmospheric nitrate at night.
As shown in Figure 3b, the more significant correlation between $NO_2 \times O_3$ and NOR under the
RH$\geq$60 % condition ($R^2$=0.534) than under the RH<60 % condition ($R^2$<0.005) at the nighttime
(19:00-6:00) during the sampling period further confirmed that the heterogeneous hydrolysis of
$N_2O_5$ on wet aerosols made a great contribution to atmospheric nocturnal nitrate under high RH
conditions.

However, the obvious increase in the mean values of NOR during the daytime (especially for

10:00-17:00) under the 30 %<RH<60 % condition (Figure 3a) indicated that additional sources
rather than the heterogeneous hydrolysis of $N_2O_5$ were responsible for the formation of nitrate. To
explore the possible formation mechanisms of nitrate in this case, the daily variations of Dust (the
sum of $Ca^{2+}$ and $Mg^{2+}$) $\times$ $NO_2$ and HONO (the main source of OH) $\times$ DR $\times$ $NO_2$, which can
represent roughly the heterogeneous reaction of $NO_2$ on the surface of mineral aerosols and the
gas-phase reaction of $NO_2$ with OH, are shown in Figure 3c and Figure 3d, respectively. The mean
values of HONO $\times$ DR $\times$ $NO_2$ during the daytime were found to be remarkably greater under the
30 %<RH<60 % condition than under the RH$\leq$30 % condition, whereas the mean values of Dust $\times$
$NO_2$ almost stayed the same under the two different RH conditions. Considering the coincident
trend of NOR and HONO $\times$ DR $\times$ $NO_2$ during the daytime (10:00-17:00) under the 30 %<RH<60 %
condition, the gas-phase reaction of $NO_2$ with OH played a key role in the diurnal formation of
nitrate at moderate RH levels with the haze pollution accumulating. It should be noted that the
mean values of HONO $\times$ DR $\times$ $NO_2$ decreased dramatically from 14:00 to 17:00 (Figure 3d),





which was not responsible for the high mean values of NOR at that time (Figure 3a). However, the
slight increase in the mean values of Dust × $NO_2$ after 14:00 was observed under the
30 %<RH<60 % condition (Figure 3c) and hence the heterogeneous reaction of $NO_2$ on the
surface of mineral aerosols was suspected to contribute to the diurnal formation of nitrate at that
time under moderate RH condition.
**3.3.2. Formation mechanism of sulfate**

Atmospheric sulfate is principally from $SO_2$ oxidation pathway, including gas-phase

reactions with OH radical or stabilized Criegee intermediates, heterogeneous-phase reactions on
the surface of particles and aqueous-phase reactions with dissolved $O_3$, $NO_2$, $H_2O_2$ and organic
peroxides, as well as autoxidation catalyzed by TMI (Cheng et al., 2016;Li et al.,
2018;Ravishankara, 1997;Shao et al., 2019;Wang et al., 2016;Xue et al., 2016;Zhang et al., 2018).
As shown in Figure 4, similar to the daily variations of NOR, the remarkable elevation of the
mean values of SOR after 14:00 under the 30 %<RH<60 % condition compared to the RH≤30 %
condition might be also ascribed to the heterogeneous reaction of $SO_2$ on the surface of mineral
aerosols. The extremely high mean values of SOR during the whole day under the RH≥60 %
condition implied that aqueous oxidation of $SO_2$ dominated the formation of sulfate during the
severe pollution episodes, which was in line with previous studies (Zhang et al., 2018;Cheng et al.,
2016). A key factor that influenced the aqueous oxidation pathways for sulfate formation has been
considered to be the aerosol pH (Guo et al., 2017;Liu et al., 2017a), varying from 4.5 to 8.5 at
different atmospheric RH and sulfate levels during the sampling period (Figure 5a) on the basis of
the ISORROPIA-II model. Considering that the aqueous-phase chemistry of sulfate formation
usually occurs in severe haze events with relatively high atmospheric RH, the aerosol pH (4.5-5.3)



under the RH≥60 % condition, which was lower than those (5.4-7.0) in the studies of Wang et al.,
(2016) and Cheng et al., (2016) but was slightly higher than those (3.0-4.9) in the studies of Liu et
al., (2017a) and Guo et al., (2017), was adopted for evaluating sulfate production in this study. In
addition, in terms of oxidants, the obvious increase in the average concentration of $NO_2$ (Figure 5b)
and the evident decrease in the average concentration of $O_3$ (Figure 5d) were observed with the
deterioration of $PM_{2.5}$ pollution. Furthermore, the average concentration of $H_2O_2$ was also found
to be extremely high (0.25 ppb) under the HP condition (Figure 5c) and was above 1 order of
magnitude higher than that (0.01 ppb) assumed by Cheng et al., (2016), which probably resulted in
the underestimation of the contribution of $H_2O_2$ to sulfate formation in the study of Cheng et al.,

(2016).

To further explore the contribution of $H_2O_2$ to sulfate production rate under the HP condition,

the parameters measured in this study (Table 2) and the same approach that was adopted by Cheng
et al., (2016) were used to calculate sulfate production. As shown in Figure 6, the relationships
between different aqueous oxidation pathways and aerosol pH in this study were found to be very
similar with those of Cheng et al., (2016). However, the contribution of $H_2O_2$ to sulfate production
rate was about a factor of 17 faster in this study (about 1.16 $\mu g\ m^{-3}\ h^{-1}$) than in the study (about
$6.95\times10^{-2}\ \mu g\ m^{-3}\ h^{-1}$) of Cheng et al., (2016), implying that the contribution of $H_2O_2$ to sulfate
formation was largely neglected. Furthermore, considering the aerosol pH calculated under the HP
condition during the sampling period, the oxidation pathway of $NO_2$ might play an insignificant
role in sulfate production rate ($8.96\times10^{-2}$-0.56 $\mu g\ m^{-3}\ h^{-1}$) and its importance proposed by the
previous studies (1.74-10.85 $\mu g\ m^{-3}\ h^{-1}$) was not necessarily expected.
**4. Conclusion**





Based on the comprehensive analysis of the pollution levels, the variation characteristics and
the formation mechanisms of the key species in $PM_{2.5}$ and the typical gaseous pollutants during
the winter haze pollution periods in Beijing, three serious haze pollution cases were obtained
during the sampling period and the SIAs formations especially nitrate and sulfate were found to
make an evident contribution to atmospheric $PM_{2.5}$ under the relatively high RH condition. The
significant correlation between $NO_2 \times O_3$ and NOR at night under the RH≥60 % condition
indicated that the heterogeneous hydrolysis of $N_2O_5$ on wet aerosols was responsible for the
nocturnal formation of nitrate under extremely high RH conditions. The more coincident trend of
NOR and HONO × DR × $NO_2$ than Dust × $NO_2$ during the daytime under the 30 %<RH<60 %
condition suggested that the gas-phase reaction of $NO_2$ with OH played a key role in the diurnal
formation of nitrate under moderate RH conditions. The extremely high mean values of SOR
during the whole day under the RH≥60 % condition could be explained by the dominant
contribution of aqueous-phase reaction of $SO_2$ to atmospheric sulfate formation during the severe
pollution episodes. According to the parameters measured in this study and the same approach that
was adopted by Cheng et al., (2016), the oxidation pathway of $H_2O_2$ rather than $NO_2$ was found to
contribute greatly to atmospheric sulfate formation.
Our results revealed that the heavy pollution events in winter usually occurred with high
concentration levels of pollutants and oxidants as well as high liquid water contents of moderately
acidic aerosols in the NCP. Thus, emission controls of $NO_x$, $SO_2$ and VOCs especially under the
extremely high RH conditions are expected to reduce largely the pollution levels of nitrate and
sulfate in northern China and even in other pollution regions of China.



*Data availability.* Data are available from the corresponding author upon request (yjmu@rcees.ac.cn)

*Author contributions.* YJM designed the experiments. PFL carried out the experiments and prepared the manuscript. CY and CYX carried out the experiments. CLZ was involved in part of the work. XS provided the meteorological data and trace gases in Beijing.

*Competing interests.* The authors declare that they have no conflict of interest.

*Acknowledgement.* This work was supported by the National research program for Key issues in air pollution control (No. DQGG0103, DQGG0209, DQGG0206), the National Natural Science Foundation of China (No. 91544211, 4127805, 41575121, 21707151), the National Key Research and Development Program of China (No. 2016YFC0202200, 2017YFC0209703, 2017YFF0108301) and Key Laboratory of Atmospheric Chemistry, China Meteorological Administration (No. 2018B03).




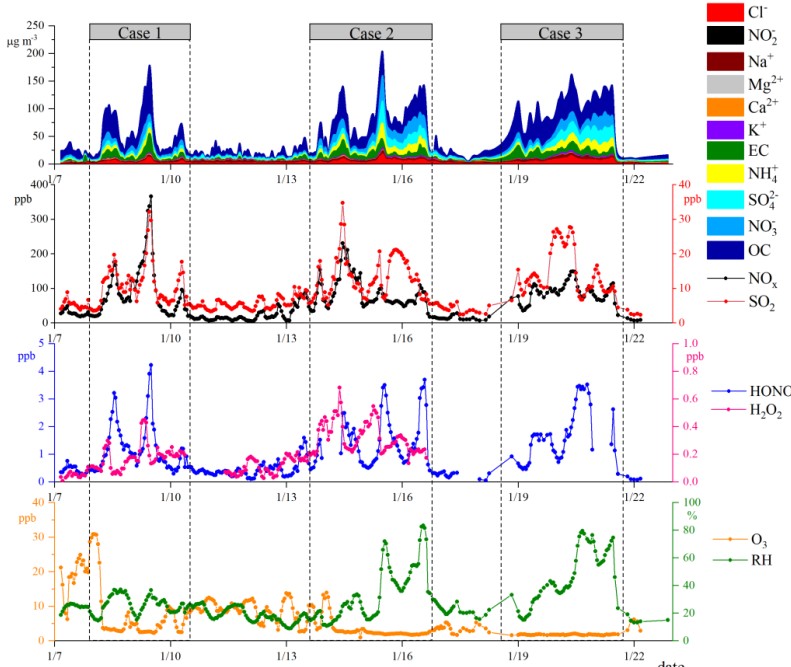


**Figure 1.** Time series of the species in PM$_{2.5}$ and typical gaseous pollutants (NO$_x$, SO$_2$, O$_3$,
HONO and H$_2$O$_2$) as well as atmospheric RH during the sampling period.

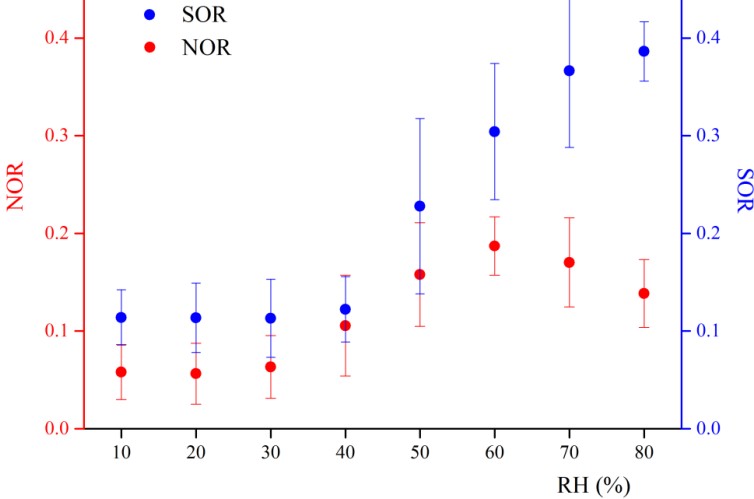


**Figure 2.** The relations between NOR as well as SOR and RH during the sampling period.




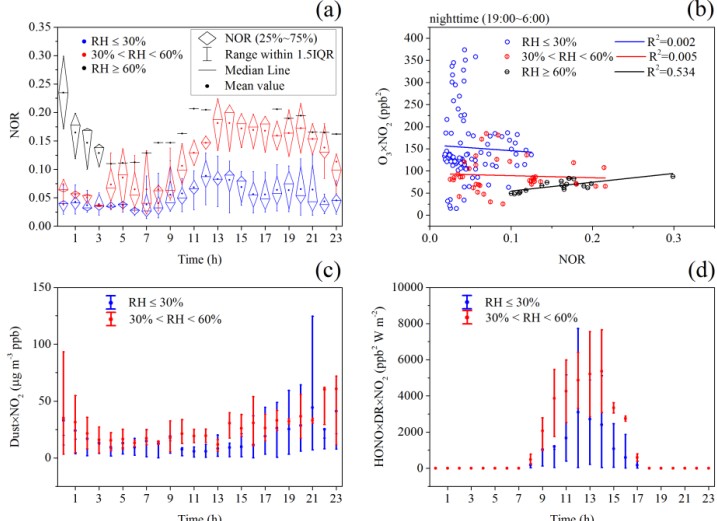


**Figure 3.** Daily variation of NOR (a), the correlation between NOR and $O_3 \times NO_2$ at the nighttime
(b), daily variations of Dust$\times NO_2$ and HONO$\times$DR$\times NO_2$ (c, d) under different atmospheric RH
conditions during the sampling period.

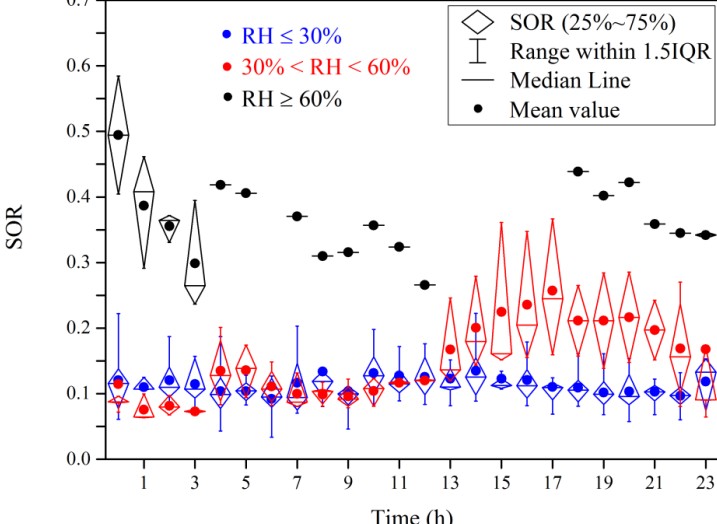


**Figure 4.** Daily variation of SOR under different atmospheric RH conditions during the sampling





period.


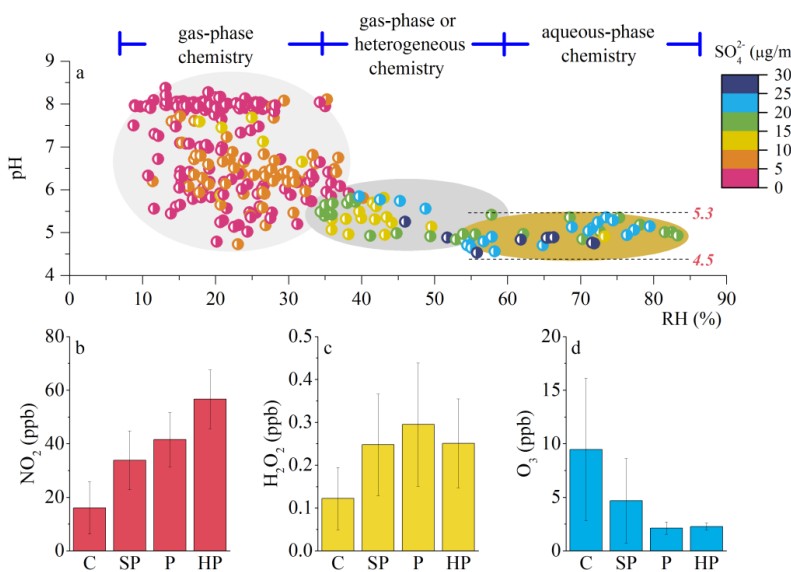


**Figure 5.** The correlations among aerosol pH, atmospheric RH and atmospheric $SO_4^{2-}$ (a), the
average concentrations of $NO_2$, $H_2O_2$ and $O_3$ (b, c, d) under different pollution conditions (clean
(C), $PM_{2.5}<35$ μg m$^{-3}$; slightly polluted (SP), 35 μg m$^{-3}<PM_{2.5}<75$ μg m$^{-3}$; polluted (P), 75 μg
m$^{-3}<PM_{2.5}<150$ μg m$^{-3}$; heavy polluted (HP), $PM_{2.5}>150$ μg m$^{-3}$) during the sampling period.

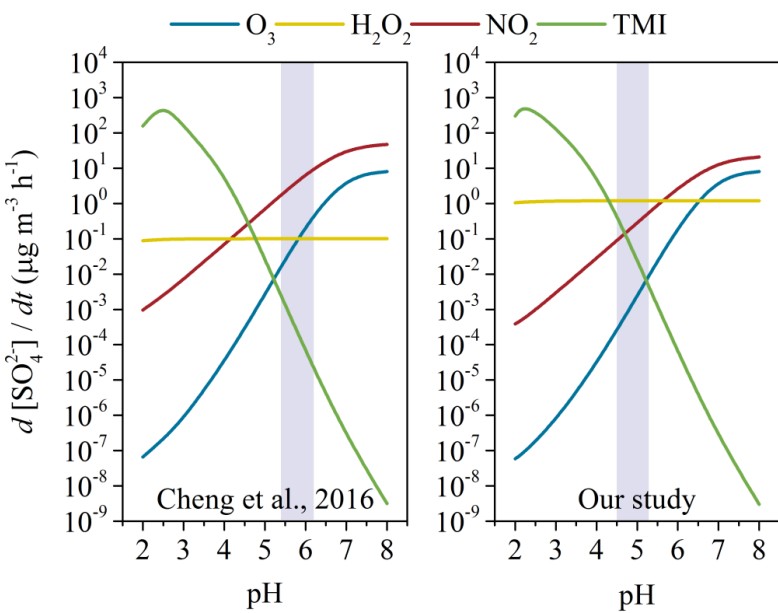






**Figure 6.** The comparison of aqueous-phase sulfate production by SO$_2$ oxidation under different

aerosol pH conditions between in the study of Cheng et al., (2016) and in this study.


**Table 1.** The average concentrations of the species in PM$_{2.5}$ (µg m$^{-3}$) and typical gaseous
pollutants (ppb) during C&SP episodes (PM$_{2.5}$<75 µg m$^{-3}$), during P&HP episodes (PM$_{2.5}$≥75 µg

m$^{-3}$) and during the whole sampling period.

| species | during C&SP episodes (n=210) | during P&HP episodes (n=108) | total (n=318) |
|---|---|---|---|
| PM$_{2.5}$ | 30.00 ± 17.79 | 113.35 ± 28.10 | 58.31 ± 45.15 |
| Na$^+$ | 2.88 ± 1.11 | 3.68 ± 1.19 | 3.15 ± 1.21 |
| Mg$^{2+}$ | 0.05 ± 0.03 | 0.08 ± 0.06 | 0.06 ± 0.04 |
| Ca$^{2+}$ | 0.52 ± 0.33 | 0.67 ± 0.48 | 0.58 ± 0.40 |
| K$^+$ | 0.81 ± 0.42 | 1.84 ± 0.73 | 1.16 ± 0.73 |
| NH$_4^+$ | 1.90 ± 1.90 | 11.52 ± 4.93 | 5.17 ± 5.62 |
| SO$_4^{2-}$ | 3.64 ± 1.87 | 14.96 ± 7.80 | 7.47 ± 7.18 |
| NO$_3^-$ | 3.44 ± 3.57 | 17.15 ± 7.36 | 8.10 ± 8.32 |
| Cl$^-$ | 1.89 ± 1.20 | 7.35 ± 2.97 | 3.73 ± 3.26 |
| NO$_2^-$ | 0.06 ± 0.08 | 0.06 ± 0.05 | 0.06 ± 0.07 |
| OC | 12.10 ± 9.25 | 43.34 ± 13.88 | 22.73 ± 18.48 |
| EC | 3.98 ± 3.42 | 12.69 ± 6.43 | 7.58 ± 6.51 |
| NO$_x$ | 39.38 ± 35.25 | 107.71 ± 58.44 | 62.59 ± 54.98 |
| NO$_2$ | 21.46 ± 13.04 | 42.81 ± 10.96 | 28.71 ± 15.98 |
| SO$_2$ | 6.99 ± 3.64 | 15.70 ± 6.55 | 9.95 ± 6.35 |
| O$_3$ | 8.01 ± 6.35 | 2.13 ± 0.56 | 6.01 ± 5.87 |
| HONO | 0.60 ± 0.43 | 1.90 ± 0.97 | 1.01 ± 0.87 |
| H$_2$O$_2$ | 0.17 ± 0.11 | 0.29 ± 0.14 | 0.20 ± 0.13 |


**Table 2.** The comparisons for parameters of sulfate production rate calculations between in the

study of Cheng et al., (2016) and in this work during the most polluted haze periods

| Parameters | This study | Cheng et al., (2016) |
|---|---|---|
| NO$_2$ | 57 ppb | 66 ppb |
| H$_2$O$_2$ | 0.25 ppb | 0.01 ppb |
| O$_3$ | 2 ppb | 1 ppb |
| SO$_2$ | 35 ppb | 40 ppb |
| Fe(III)[a] | 18 ng m$^{-3}$ | 18 ng m$^{-3}$ |
| Mn(II)[a] | 42 ng m$^{-3}$ | 42 ng m$^{-3}$ |
| ALWC | 146 µg m$^{-3}$ | 300 µg m$^{-3}$ |
| Aerosol droplet radius (R)[a] | 0.15 µm | 0.15 µm |
| Temperature | 270 K | 271 K |
| pH | 4.5-5.3 | 5.4-6.2 |

[a]: both the concentrations of Fe(III) and Mn(II) and aerosol droplet radius were not measured in
this study and were derived from Cheng et al., (2016).

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
