# Peer review of "Formation mechanisms of atmospheric nitrate and sulfate during the"

_Atmospheric Chemistry and Physics, 2019_

## Referee Comment (RC1) · Anonymous Referee #1 · 11 Dec 2019

This study investigates the pollution characteristics and formation mechanisms of sulfate and nitrate during the winter haze pollution periods in Beijing in 2016 based on the field observations, which is helpful for us to understand the winter haze formation in China and better control it. However, some discussions are confusing. This paper cannot be accepted before the authors have addressed the following comments.

**1. Line 257-258:** The variation of $NO_2$ should be given in Figure 1, because the discussion in this study is based on $NO_2$, not $NO_X$. Are the concentrations of $SO_2$ a factor of 5 lower than the concentrations of $NO_2$? Based on Table 1, it's not true.

**2. Line 262:** NOR is commonly defined as "$NOR = NO_3^- / (NO_3^- + NO_2)$" in the previous studies. Why is "$NOR = NO_3^- / (NO_3^- + NO_x)$" used in this study? We know that NOx is usually much higher than $NO_2$, especially at night. Are the discussion and results different if you use "$NOR = NO_3^- / (NO_3^- + NO_2)$" in this study? If you use "$NOR = NO_3^- / (NO_3^- + NO_x)$", why do you use "$NO_2*O_3$", "$Dust*NO_2$" and "$HONO*DR*NO_2$" in the following discussion, rather than "$NO_x*O_3$", "$Dust*NO_x$" and "$HONO*DR*NO_x$"?

**3. Line 270-271:** Why is the reduction of NOR due to the deliquescence of nitrate? Based on the reference you list, the deliquescence can change aerosol particle size distribution, but not decrease the nitrate concentration.

**4. Section 3.3.1:** The discussion about the nitrate formation is not convincing and more analysis is needed.
**(1) Line 295-299**: Is the correlation analysis in figure 3b proper to investigate the contribution of heterogeneous hydrolysis of $N_2O_5$? Why does a negative correlation exist between NOR and $NO_2*O_3$ under the RH<60 % condition? It means that the heterogeneous hydrolysis of $N_2O_5$ is not important under the RH<60 % condition? If the authors use a similar figure with Figure3b to analyze the correlation between NOR with $Dust*NO_2$ and $HONO*DR*NO_2$, what are the conclusions?
**(2) Figure 3a**: Why does NOR decrease obviously during 0:00-4:00 under the RH>60 % condition?
**(3) Line 296-297:** why is the nighttime defined as 19:00-6:00? Seeing from Figure 3d, DR is nearly zero during 18:00-7:00.
**(4) Line 300-302:** A similar NOR increase during the daytime can be seen under the RH>60 % condition in Figure 3a. Why are the hourly variations of $Dust*NO_2$ and $HONO*DR*NO_2$ under the RH>60 % condition not included in Figure 3c-d. Seeing from the abstract (Line 27-30) and conclusion part (Line 365-368), the authors seem to conclude the gas-phase reaction of $NO_2$ with OH plays a key role just under moderate RH conditions. How about under the RH>60% or RH<30% condition?

**5. Section 3.3.2**

**(1) Line 325-328:** Why is the heterogeneous reaction of $SO_2$ on the surface of mineral aerosols not important before 14:00?

**(2) Figure 4**: Why does SOR decrease obviously during 0:00-4:00 under the RH>60 % condition?

**(3) Figure 4**: Under 30%<RH<60%, why is the SOR during 13:00-23:00 much higher than that in other hours? We know that RH is commonly high at night, for example during 0:00-5:00.

---

## Referee Comment (RC2) · Anonymous Referee #2 · 12 Dec 2019

This study focused on the formation mechanisms of nitrate and sulfate in Beijing, especially the different mechanisms under various RH conditions. The heterogeneous hydrolysis of N2O5 was responsible for the nocturnal formation of nitrate at extremely high RH levels (RH>60%), while homogeneous reaction between NO2 and OH radical dominated the formation under moderate condition (30%<RH<60%). For SO42-, aqueous reaction between SO2 and H2O2 attributed to its formation under high RH condition. The target of this study is meaningful to understanding the formation mechanism of nitrate and sulfate in real atmosphere. There are several questions not very clear.

[Figure]

Comments: 1. Please give a brief description of NOR and SOR in abstract. 2. Did NOR and SOR represent the secondary formation of $NO_3^-$ and $SO_4^{2-}$, respectively? Actually, when NOx and SO2 reached zero, the value of NOR and SOR were closed to the maximum. If NOR and SOR represent the secondary formation of $NO_3^-$ and $SO_4^{2-}$, secondary formation of $NO_3^-$ and $SO_4^{2-}$ showed up with low concentration of NOx and SO2. This result is confusing. 3. The authors mentioned that "The reduction of NOR might be due to the deliquescence of nitrate at atmospheric RH around 60 %" at line 270-271. However, the deliquescence of nitrate would not reduce the nitrate in particle but change its phase state. RH has been validated to affect the heterogeneous reaction of NOx and HONO, which may result in the reduction of nitrate at high RH condition. 4. One N2O5 could be generated by two NO2 reacting with one O3. Hence, is it more suitable to use $[NO2]^2 \times [O3]$ rather than $[NO2] \times [O3]$ for representing the heterogeneous hydrolysis of N2O5 to atmospheric nitrate at night? 5. Though HONO is a main source OH, the diurnal variation of HONO may be different from OH radical. Have the author ever analyzed the correlation between $DR \times NO2$ and NOR? Because the diurnal variation of OH radical should be highly correctly with radiation.

---

## Author Comment (AC1) · 11 Feb 2020

A point-by-point response to the reviews

We are very thankful to two reviewers for your valuable comments and thoughtful suggestions. The followings are our responses to your comments. The comments of the reviewers are shown in black, our responses to the comments are presented in blue, and the new or modified texts are provided in blue in *italics*.

**Response to Reviewer #1**

**Comment 1:** This study investigates the pollution characteristics and formation mechanisms of sulfate and nitrate during the winter haze pollution periods in Beijing in 2016 based on the field observations, which is helpful for us to understand the winter haze formation in China and better control it. However, some discussions are confusing. This paper cannot be accepted before the authors have addressed the following comments.

**Answer:** Thank you for your pertinent evaluation of our work. The followings are our responses to your comments.

**Comment 2:** Line 257-258: The variation of $NO_2$ should be given in Figure 1, because the discussion in this study is based on $NO_2$, not $NO_x$. Are the concentrations of $SO_2$ a factor of 5 lower than the concentrations of $NO_2$? Based on Table 1, it's not true.

**Answer:** The time series of $NO_x$ have been replaced with the variations of $NO_2$ in the revised Figure 1 (Figure R1). On the basis of Figure R1 and Table 1, the concentrations of $SO_2$ were about a factor of 5-6 lower than those of $NO_x$, but were approximately a factor of 3 lower than those of $NO_2$. This sentence has been changed in the revised manuscript as following:

[Figure]

**Figure R1.** Time series of the species in $PM_{2.5}$ and typical gaseous pollutants ($NO_2$, $SO_2$, $O_3$, HONO and $H_2O_2$) as well as atmospheric RH during the sampling period.

*"Although the concentrations of $SO_2$ were obviously lower than the concentrations of $NO_2$ in both Case 2 and Case 3 (Figure 1 and Table 1), ..."*

**Comment 3:** Line 262: NOR is commonly defined as "NOR = $NO_3^-$/ ($NO_3^-$+$NO_2$)" in the previous studies. Why is "NOR = $NO_3^-$/ ($NO_3^-$+$NO_x$)" used in this study? We know that $NO_x$ is usually much higher than $NO_2$, especially at night. Are the discussion and results different if you use "NOR = $NO_3^-$/ ($NO_3^-$+$NO_2$)" in this study? If you use "NOR = $NO_3^-$/ ($NO_3^-$+$NO_2$)", why do you use "$NO_2 \times O_3$", "Dust $\times NO_2$" and "HONO $\times$ DR $\times NO_2$" in the following discussion, rather than "$NO_x \times O_3$", "Dust $\times NO_x$" and "HONO $\times$ DR $\times NO_x$"?

**Answer:** We are very sorry for our incorrect writing of the NOR formula. In fact, "NOR = $NO_3^-$/ ($NO_3^-$+$NO_2$)" rather than "NOR = $NO_3^-$/ ($NO_3^-$+$NO_x$)" was used in this study. As NO concentration usually accounted for relatively high fraction to that of $NO_x$ in winter of Beijing city, especially during the morning and evening rushing hours, NOR calculated based on $NO_2$ was obviously higher than that based on $NO_x$ (Figure R2). Because atmospheric nitrate is formed through the oxidation of $NO_2$ via gas-phase, heterogeneous and aqueous-phase reactions, "NOR = $NO_3^-$/ ($NO_3^-$+$NO_2$)" might reflect nitrogen oxidation ratio more accurately than "NOR = $NO_3^-$/ ($NO_3^-$+$NO_x$)". The mistake has been corrected in the revised manuscript as following:

[Figure]

**Figure R2.** The comparison of the time series of NOR calculated based on $NO_2$ and $NO_x$ during the sampling period.

*"To further investigate the pollution characteristics of nitrate and sulfate during the serious pollution episodes, the relations between NOR (NOR = $NO_3^-$ / ($NO_3^-$+$NO_2$)) as well as SOR (SOR = $SO_4^{2-}$ / ($SO_4^{2-}$+$SO_2$)) and RH are shown in Figure 2."*

**Comment 4:** Line 270-271: Why is the reduction of NOR due to the deliquescence of nitrate? Based on the reference you list, the deliquescence can change aerosol particle size distribution, but not decrease the nitrate concentration.

**Answer:** According to the Comment 4 from the Reviewer #2, the reduction of NOR might be ascribed to the suppressed heterogeneous reactions of $NO_2$ to nitrate formation under high RH condition (Tang et al., 2017). The heterogeneous reactions of $NO_2$ on particle surface have been found to be dependent on atmospheric RH due to the competition of water for surface reactive sites of particles (Ponczek et al., 2019), and thus relatively fast nitrate formation usually occurs when RH is below a certain value. This sentence has been rephrased in the revised manuscript as following:

*"the variation trend of NOR slowly decreased whereas the variation trend of SOR significantly increased when atmospheric RH was above 60 %, which was very similar with the previous studies (Sun et al., 2013; Zheng et al., 2015b). Considering that the heterogeneous reactions of $NO_2$ on particle surface were dependent on atmospheric RH due to the competition of water for surface reactive sites of particles (Ponczek et al., 2019), the slow reduction of NOR might be due to the suppressed heterogeneous reaction of $NO_2$ to nitrate formation under high RH condition (Tang et al., 2017), while the elevation of SOR revealed the dominant contribution of the aqueous-phase reaction to sulfate formation."*

**Comment 5:** Section 3.3.1: The discussion about the nitrate formation is not convincing and more analysis is needed.
(1) Line 295-299: Is the correlation analysis in Figure 3b proper to investigate the contribution of heterogeneous hydrolysis of $N_2O_5$? Why does a negative correlation exist between NOR and $NO_2$ × $O_3$ under the RH<60 % condition? It means that the heterogeneous hydrolysis of $N_2O_5$ is not important under the RH<60 % condition? If the authors use a similar figure with Figure 3b to analyze the correlation between NOR with Dust × $NO_2$ and HONO × DR × $NO_2$, what are the conclusion?

**Answer:** Considering that one molecule of $N_2O_5$ could be generated by two molecule of $NO_2$ reacting with one molecule of $O_3$, perhaps it's more proper to use the correlation between NOR and $[NO_2]^2$ × $[O_3]$ rather than $[NO_2]$ × $[O_3]$ for representing the contribution of the heterogeneous hydrolysis of $N_2O_5$ to atmospheric nitrate at night (the Comment 5 from the Reviewer #2). The correlations between NOR and $[NO_2]^2$ × $[O_3]$ at the nighttime (redefined as 18:00-7:00) under different RH conditions are shown in Figure R3. It's evident that the variations of $[NO_2]^2$ × $[O_3]$ were all positively correlated with NOR under the three different RH conditions, and their correlation under the RH ≥ 60% condition ($R^2$ = 0.552) was significantly stronger than those under the RH < 60% condition ($R^2$ ≤ 0.181). It has been acknowledged that the correlation between two species means the impact of changes in one species on another. The stronger the correlation is, the

greater the impact is. Therefore, the positive correlations between NOR and $[NO_2]^2 \times [O_3]$ indicated that the heterogeneous hydrolysis of $N_2O_5$ could contribute to the formation of atmospheric nitrate at the nighttime under different RH conditions. The significantly stronger correlations between NOR and $[NO_2]^2 \times [O_3]$ under the RH $\geq$ 60% condition than under the RH < 60% condition revealed that the heterogeneous hydrolysis of $N_2O_5$ made a remarkable contribution to atmospheric nitrate at the nighttime under high RH condition. Additionally, the obviously lower slope of the correlation between NOR and $[NO_2]^2 \times [O_3]$ under the RH $\geq$ 60% condition (slope = 11691) than under the RH < 60% condition (slope $\geq$ 17399) also suggested that the formation of atmospheric nitrate at the nighttime under high RH condition was more sensitive to the pathway of $N_2O_5$. These sentences have been modified in the revised manuscript as stated above.

It should be noted that the correlations between NOR and $[NO_2]^2 \times [O_3]$ under the three different RH conditions were analyzed for verifying under which RH condition the heterogeneous hydrolysis of $N_2O_5$ made a remarkable contribution to atmospheric nitrate at the nighttime, while the daily variations of $[Dust] \times [NO_2]$ and $[HONO] \times [DR] \times [NO_2]$ under the 30% < RH < 60% condition were compared for exploring which reaction could play an important role in atmospheric nitrate at the daytime under moderate RH condition. Therefore, it may be lack of the purpose to analyze the correlations between NOR and $[Dust] \times [NO_2]$ as well as $[HONO] \times [DR] \times [NO_2]$ by using the similar method of Figure R3.

[Figure]

**Figure R3.** The correlations between NOR and $[NO_2]^2 \times [O_3]$ at the nighttime under different RH conditions.

(2) Figure 3a: Why does NOR decrease obviously during 0:00-4:00 under the RH>60 % condition?

**Answer:** As mentioned above, the heterogeneous hydrolysis of $N_2O_5$ was found to make a remarkable contribution to atmospheric nitrate at the nighttime under the RH ≥ 60% condition due to the strong correlation between NOR and $[NO_2]^2 \times [O_3]$. Thus, the obvious reduction of the NOR values during 0:00-3:00 under the RH ≥ 60% condition was mainly ascribed to the decrease in the concentration levels of $[NO_2]^2 \times [O_3]$ (Figure R4).

[Figure]

**Figure R4.** The variations of $[NO_2]^2 \times [O_3]$ and $[SO_2] \times [H_2O_2]$ during 0:00-3:00 under the RH ≥ 60% condition during the sampling period.

(3) Line 296-297: why is the nighttime defined as 19:00-6:00? Seeing from Figure 3d, DR is nearly zero during 18:00-7:00.

**Answer:** Thank you for your suggestion. The nighttime has been redefined as 18:00-7:00 and the associated figures have been changed in the revised manuscript.

(4) Line 300-302: A similar NOR increase during the daytime can be seen under the RH>60 % condition in Figure 3a. Why are the hourly variations of Dust × $NO_2$ and HONO × DR × $NO_2$ under the RH>60 % condition not included in Figure 3c-d. Seeing from the abstract (Line 27-30) and conclusion part (Line 365-368), the authors seem to conclude the gas-phase reaction of $NO_2$ with OH plays a key role just under moderate RH conditions. How about under the RH>60 % or RH<30 % condition?

**Answer:** The relatively high atmospheric RH (RH > 60%) usually occurred at the nighttime

during the sampling period (Figure R4), and hence it was difficult by using [HONO] × [DR] × [NO₂] to conclude whether the gas-phase reaction of NO₂ with OH played a key role under the RH > 60% condition. Furthermore, because the NOR values under the RH ≤ 30% condition were almost less than 0.1 (Figure 2 in the revised manuscript) which reflected no occurrence of secondary formation of nitrate (Gao et al., 2011; Zhang et al., 2018), it might be not necessary to discuss the formation of nitrate under the RH ≤ 30% condition.

**Comment 6:** Section 3.3.2:
(1) Line 325-328: Why is the heterogeneous reaction of SO₂ on the surface of mineral aerosols not important before 14:00?

**Answer:** Atmospheric sulfate has been reported to come mainly from primary source emissions when the SOR is less than 0.1 (Gao et al., 2011; Zhang et al., 2018). Considering that the mean values of SOR before 14:00 both under the 30% < RH < 60% and RH ≤ 30% conditions were almost close to 0.1, secondary formation of SO₂ including the gas-phase reaction and heterogeneous reaction could be ignored before 14:00. To avoid possible confusing understanding for readers, this sentence has been rephrased in the revised manuscript as following:

*"As shown in Figure 4, similar to the daily variations of NOR, the mean values of SOR were found to elevated remarkably under the 30%<RH<60% condition compared to the RH≤30% condition, especially during 14:00-22:00, which might be mainly ascribed to the enhanced gas-phase reaction and the heterogeneous reaction of SO₂ involving aerosol liquid water under the relatively high RH condition."*

(2) Figure 4: Why does SOR decrease obviously during 0:00-4:00 under the RH>60 % condition?

**Answer:** Because the oxidation of SO₂ through the aqueous-phase reaction of H₂O₂ was found to contribute mainly to sulfate formation under the high RH condition, the depletion of the oxidant and the precursor ([SO₂] × [H₂O₂]) during 0:00-3:00 was suspected to result in the obvious decrease of SOR under the RH ≥ 60% condition (Figure R4).

(3) Figure 4: Under 30 %<RH<60 %, why is the SOR during 13:00-23:00 much higher than that in other hours? We know that RH is commonly high at night, for example during 0:00-5:00.

**Answer:** Yes. atmospheric RH is indeed a key factor for influencing sulfate formation and commonly high at night. Except for atmospheric RH, the concentrations of the precursors such as SO₂ could also play a vital role in the formation of sulfate, and then affected SOR value. Therefore, the much higher mean values of SOR during 13:00-23:00 than those in other hours might be mainly attributed to the relatively high concentrations of SO₂ during 13:00-23:00 under 30 %<RH<60 % condition (Figure R5).

[Figure]

**Figure R5.** The daily variation of $SO_2$ under the 30% < RH <60% condition during the sampling period

**Response to Reviewer #2**

**Comment 1:** This study focused on the formation mechanisms of nitrate and sulfate in Beijing, especially the different mechanisms under various RH conditions. The heterogeneous hydrolysis of $N_2O_5$ was responsible for the nocturnal formation of nitrate at extremely high RH levels (RH>60 %), while homogeneous reaction between $NO_2$ and OH radical dominated the formation under moderate condition (30 %<RH<60 %). For $SO_4^{2-}$, aqueous reaction between $SO_2$ and $H_2O_2$ attributed to its formation under high RH condition. The target of this study is meaningful to understanding the formation mechanism of nitrate and sulfate in real atmosphere. There are several questions not very clear.

**Answer:** Thank you for your valuable evaluation of our work. The followings are our responses to your comments.

**Comment 2:** Please give a brief description of NOR and SOR in abstract.

**Answer:** the NOR and SOR formulas have been added in the revised abstract.

**Comment 3:** Did NOR and SOR represent the secondary formation of $NO_3^-$ and $SO_4^{2-}$, respectively? Actually, when $NO_x$ and $SO_2$ reached zero, the value of NOR and SOR were closed to the maximum. If NOR and SOR represent the secondary formation of $NO_3^-$ and $SO_4^{2-}$,

secondary formation of $NO_3^-$ and $SO_4^{2-}$ showed up with low concentration of $NO_x$ and $SO_2$. This result is confusing.

**Answer:** NOR (NOR = $NO_3^-$/ ($NO_3^-$+$NO_2$)) and SOR (SOR = $SO_4^{2-}$/ ($SO_4^{2-}$+$SO_2$)) didn't represent the secondary formation of $NO_3^-$ and $SO_4^{2-}$, but could reflect their formation potentials to a certain degree due to the ratios counteracted the air diffusion effect on their concentrations, and thus NOR and SOR have been widely used to estimate the secondary formation of $NO_3^-$ and $SO_4^{2-}$, respectively (Zheng et al., 2015). Yes, the NOR and SOR would be close to the maximal values if $NO_2$ and $SO_2$ reached zero. Actually, $NO_2$ and $SO_2$ are ubiquitous trace gases in the atmosphere, it is impossible that their concentrations reached zero. The values of NOR and SOR mainly depend on the conversion efficiencies of $NO_2$ and $SO_2$ to nitrate and sulfate through various atmospheric chemical reactions, rather than the concentrations of $NO_2$ and $SO_2$, because the concentrations of $NO_2$, $SO_2$, nitrate and sulfate usually have the similar variation trends which are mainly governed by meteorological conditions and boundary layer heights as well.

**Comment 4:** The authors mentioned that "The reduction of NOR might be due to the deliquescence of nitrate at atmospheric RH around 60 %" at line 270-271. However, the deliquescence of nitrate would not reduce the nitrate in particle but change its phase state. RH has been validated to affect the heterogeneous reaction of $NO_x$ and HONO, which may result in the reduction of nitrate at high RH condition.

**Answer:** Thank you for your valuable comment. According to your suggestion, this sentence has been rephrased in the revised manuscript as following:

*"the variation trend of NOR slowly decreased whereas the variation trend of SOR significantly increased when atmospheric RH was above 60 %, which was very similar with the previous studies (Sun et al., 2013; Zheng et al., 2015b). Considering that the heterogeneous reactions of $NO_2$ on particle surface were dependent on atmospheric RH due to the competition of water for surface reactive sites of particles (Ponczek et al., 2019), the slow reduction of NOR might be due to the suppressed heterogeneous reaction of $NO_2$ to nitrate formation under high RH condition (Tang et al., 2017), while the elevation of SOR revealed the dominant contribution of the aqueous-phase reaction to sulfate formation."*

**Comment 5:** One $N_2O_5$ could be generated by two $NO_2$ reacting with one $O_3$. Hence, is it more suitable to use $[NO_2]^2 \times [O_3]$ rather than $[NO_2] \times [O_3]$ for representing the heterogeneous hydrolysis of $N_2O_5$ to atmospheric nitrate at night?

**Answer:** Thank you for your valuable suggestion. Relevant figure (Figure R3) and sentences have been modified accordingly in the revised manuscript as following:

*"…Therefore, the correlation between $[NO_2]^2 \times [O_3]$ and NOR can represent roughly the contribution of the heterogeneous hydrolysis of $N_2O_5$ to atmospheric nitrate at night…"*

**Comment 6:** Though HONO is a main source OH, the diurnal variation of HONO may be

different from OH radical. Have the author ever analyzed the correlation between DR × NO$_2$ and NOR? Because the diurnal variation of OH radical should be highly correctly with radiation.

**Answer:** Because the photolysis of atmospheric HONO has been considered as the dominant OH source in polluted areas (Wang et al., 2017), the nitrate formation rate through the gas-phase reaction of NO$_2$ with OH radicals could be reflected by the product of [HONO] × [DR] × [NO$_2$]. The evident difference for the diurnal variations between the products of [HONO] × [DR] × [NO$_2$] and [DR] × [NO$_2$] implied that the relatively high HONO concentrations in the morning under the 30% < RH 60% condition played a significant role in nitrate formation (Figure R6), which was in line with the variations of NOR (Figure 3 in the revised manuscript). Therefore, it may be more proper to use [HONO] × [DR] × [NO$_2$] rather than [DR] × [NO$_2$] for representing the gas-phase reaction of NO$_2$ with OH.

[Figure]

**Figure R6.** The comparison of the daily variations of [HONO] × [DR] × [NO$_2$] and [DR] × [NO$_2$] under the RH ≤ 30% condition and under the 30% < RH < 60% condition during the sampling period.

**References**

Gao, X., Yang, L., Cheng, S., Gao, R., Zhou, Y., Xue, L., Shou, Y., Wang, J., Wang, X., Nie, W., Xu, P., and Wang, W.: Semi-continuous measurement of water-soluble ions in PM2.5 in Jinan, China: Temporal variations and source apportionments, Atmospheric Environment, 45, 6048-6056, 10.1016/j.atmosenv.2011.07.041, 2011.

Ponczek, M., Hayeck, N., Emmelin, C., and George, C.: Heterogeneous photochemistry of dicarboxylic acids on mineral dust, Atmospheric Environment, 212, 262-271, 10.1016/j.atmosenv.2019.05.032, 2019.

Tang, M., Huang, X., Lu, K., Ge, M., Li, Y., Cheng, P., Zhu, T., Ding, A., Zhang, Y., Gligorovski,

S., Song, W., Ding, X., Bi, X., and Wang, X.: Heterogeneous reactions of mineral dust aerosol: implications for tropospheric oxidation capacity, Atmospheric Chemistry and Physics, 17, 11727-11777, 10.5194/acp-17-11727-2017, 2017.

Wang, J., Zhang, X., Guo, J., Wang, Z., and Zhang, M.: Observation of nitrous acid (HONO) in Beijing, China: Seasonal variation, nocturnal formation and daytime budget, The Science of the total environment, 587-588, 350-359, 10.1016/j.scitotenv.2017.02.159, 2017.

Zhang, R., Sun, X., Shi, A., Huang, Y., Yan, J., Nie, T., Yan, X., and Li, X.: Secondary inorganic aerosols formation during haze episodes at an urban site in Beijing, China, Atmospheric Environment, 177, 275-282, 10.1016/j.atmosenv.2017.12.031, 2018.

Zheng, G. J., Duan, F. K., Su, H., Ma, Y. L., Cheng, Y., Zheng, B., Zhang, Q., Huang, T., Kimoto, T., Chang, D., Pöschl, U., Cheng, Y. F., and He, K. B.: Exploring the severe winter haze in Beijing: the impact of synoptic weather, regional transport and heterogeneous reactions, Atmospheric Chemistry and Physics, 15, 2969-2983, 10.5194/acp-15-2969-2015, 2015.

---

## Author Response (AR2)

**A point-by-point response to the reviews**

We appreciate the editor and reviewers very much for your positive and constructive comments and suggestions on our manuscript. The followings are our responses to your comments. The comments of the reviewers are shown in black, our responses to the comments are presented in blue, and the new or modified texts are provided in *italics*.

**Response to Reviewer #2**

**Comment 1:** The revised manuscript has been greatly improved. There is still one question that I want to ask. After revising, the correlation between $[NO_2]^2 \times [O_3]$ and NOR was plotted in Fig. 3b. However, the correlation of them under RH < 60% condition is very weak. Hence, I think maybe it is not necessary to address the significance of the heterogeneous hydrolysis of $N_2O_5$ under RH < 60%. Discussion on the results at RH > 60% in detail is enough.

**Answer:** Thank you for your positive evaluation of our work and your valuable suggestion. These relevant sentences have been modified and deleted in the revised manuscript as following:

*"As shown in Figure 3b, although the variations of $[NO_2]^2 \times [O_3]$ at the nighttime (18:00-7:00) were all positively correlated with NOR under the three different RH conditions, and their correlation under the RH $\geq$ 60% condition ($R^2 = 0.552$) was significantly stronger than those under the RH < 60% condition ($R^2 \leq 0.181$). It has been acknowledged that the correlation between two species means the impact of changes in one species on another. The stronger the correlation is, the greater the impact is. Therefore, the positive correlations between NOR and $[NO_2]^2 \times [O_3]$ indicated that the heterogeneous hydrolysis of $N_2O_5$ could contribute to the formation of atmospheric nitrate at the nighttime under different RH conditions. The the significantly stronger correlations between NOR and $[NO_2]^2 \times [O_3]$ under the RH $\geq$ 60% condition than under the RH < 60% condition revealed that the heterogeneous hydrolysis of $N_2O_5$ made a remarkable contribution to atmospheric nitrate at the nighttime under high RH condition."*

**Comment 2:** In addition, line 599-601 "Considering that the formation of atmospheric $NO_3$ radical is mainly via the oxidation of $NO_2$ by $O_3$, the heterogeneous hydrolysis of $N_2O_5$ occurs only at high $O_3$ and $NO_2$ levels during the nighttime (He et al., 2018; Wang et al., 2018b)." This sentence is a little incompatible with the result in Fig 3b. Because in Fig. 3b, the highest concentration of $NO_2$ and $O_3$ was shown at RH < 30% and the lowest concentration of them was shown at RH > 60%. However, the contribution of heterogeneous hydrolysis of $N_2O_5$ to nitrate was not consistent with that result. Hence, it is better to rethink the way of expression.

**Answer:** Special thanks to you for your good comment. This sentence has been changed in the revised manuscript as following:

*"Considering that the formation of atmospheric $NO_3$ radical is mainly generated via the oxidation of $NO_2$ by $O_3$, the relatively high $O_3$ and $NO_2$ levels could be in favor of the heterogeneous hydrolysisformation of $N_2O_5$ occurs only at high $O_3$ and $NO_2$ levels during the nighttime (He et al.,*

*2018;Wang et al., 2018b), and hence the correlation between [NO$_2$]$^2$ × [O$_3$] and NOR can represent roughly the contribution of the heterogeneous hydrolysis of N$_2$O$_5$ to atmospheric nitrate at night."*